# Investigation of the Morphology and Electrical Properties of Graphene Used in the Development of Biosensors for Detection of Influenza Viruses

**DOI:** 10.3390/bios12010008

**Published:** 2021-12-23

**Authors:** Natalia M. Shmidt, Alexander S. Usikov, Evgeniia I. Shabunina, Alexey V. Nashchekin, Ekaterina V. Gushchina, Ilya A. Eliseev, Vasily N. Petrov, Mikhail V. Puzyk, Oleg V. Avdeev, Sergey A. Klotchenko, Sergey P. Lebedev, Elena M. Tanklevskaya, Yuri N. Makarov, Alexander A. Lebedev, Andrey V. Vasin

**Affiliations:** 1Ioffe Institute, 26 Politekhnicheskaya, 194021 St. Petersburg, Russia; Ekaterina.Gushchina@mail.ioffe.ru (E.V.G.); Ilya.Eliseyev@mail.ioffe.ru (I.A.E.); krishkis@i.ua (V.N.P.); lebedev.sergey@mail.ioffe.ru (S.P.L.); elena.tanklevskaya@gmail.com (E.M.T.); shura.lebe@mail.ioffe.ru (A.A.L.); 2Nitride Crystals Group, 27 Pr. Engelsa, 194223 St. Petersburg, Russia; alexander.usikov@nitride-crystals.com (A.S.U.); Oleg.Avdeev@n-crystals.spb.ru (O.V.A.); yuri.makarov@nitride-crystals.com (Y.N.M.); 3School of Computer Technologies and Control, ITMO University, Lomonosova St. 9, 191002 St. Petersburg, Russia; 4Nitride Crystals, Inc., 181 E Industry Ct., Ste. B, Deer Park, NY 11729, USA; 5Faculty of Chemistry, Herzen State Pedagogical University, 48 Moika Embankment, 191186 St. Petersburg, Russia; puzyk@mail.ru; 6Smorodintsev Research Institute of Influenza, Prof. Popov St. 15/17, 197376 St. Petersburg, Russia; fosfatik@mail.ru (S.A.K.); vasin_av@spbstu.ru (A.V.V.); 7Institute of Biomedical Systems and Biotechnologies, Peter the Great St. Petersburg Polytechnic University, 29 Polytechnicheskaya Str., 195251 St. Petersburg, Russia

**Keywords:** graphene, biosensors, influenza viruses

## Abstract

In this study, we discuss the mechanisms behind changes in the conductivity, low-frequency noise, and surface morphology of biosensor chips based on graphene films on SiC substrates during the main stages of the creation of biosensors for detecting influenza viruses. The formation of phenylamine groups and a change in graphene nano-arrangement during functionalization causes an increase in defectiveness and conductivity. Functionalization leads to the formation of large hexagonal honeycomb-like defects up to 500 nm, the concentration of which is affected by the number of bilayer or multilayer inclusions in graphene. The chips fabricated allowed us to detect the influenza viruses in a concentration range of 10^−16^ g/mL to 10^−10^ g/mL in PBS (phosphate buffered saline). Atomic force microscopy (AFM) and scanning electron microscopy (SEM) revealed that these defects are responsible for the inhomogeneous aggregation of antibodies and influenza viruses over the functionalized graphene surface. Non-uniform aggregation is responsible for a weak non-linear logarithmic dependence of the biosensor response versus the virus concentration in PBS. This feature of graphene nano-arrangement affects the reliability of detection of extremely low virus concentrations at the early stages of disease.

## 1. Introduction

The rapid spread the coronavirus disease 2019 (COVID-19) during the pandemic and regularly occurring epidemics of influenza, which have killed hundreds of millions of people and caused significant damage to the global economy, have shown the need to create highly sensitive biosensors that allow quick (within minutes) detection of extremely low concentrations of antigens (viruses) at early stages in these diseases. The fabrication of such biosensors would make it possible to find out the mechanism of the spread of COVID-19. Recent reports have proposed graphene as a prospective material for these biosensors [1,2,3,4,5,6] thanks to its unusual physical properties, which are different from sensors’ 3D bulk counterparts. Graphene is a two-dimensional single atomic layer of sp^2^ bonded carbon atoms arranged in a honeycomb lattice. Two-dimensional (2D) materials are characterized by their strong interplanar bonding but weak interplanar interaction. Interfaces between neighboring 2D layers or between 2D overlayers and substrate surfaces provide intriguing confined spaces for chemical processes, which have stimulated a new area of “chemistry under 2D cover”. The interaction between electrons and the honeycomb carbon lattice causes the electrons to behave as massless fermions, which give rise to novel physical phenomena such as anomalous room temperature quantum hall effect, extraordinarily high carrier mobility, high surface area per unit volume, and a low-noise [7,8,9,10]. Thus, graphene is a very promising material for the manufacture of various types of sensors. However, the combination of these properties leads to the fact that even a minimal amount of impurity in the graphene surface can noticeably change the conductivity of the graphene film.

When it comes to the registration of influenza viruses and the severe acute respiratory syndrome coronavirus 2 (SARS-CoV-2) that causes COVID-19, several reports have shown the great promise of the application of chips based on graphene films with two contacts, i.e., graphene resistances [1,2]. The main stages of creating biosensors for the influenza and SARS-CoV-2 viruses are similar. Therefore, influenza viruses can act as an affordable, safe, and better-studied model material. All studies of graphene-based biosensors are aimed mostly at creating a design of the sensor topology that would provide a quick and reliable response to the presence of antigens on the graphene surface.

Significant improvements in the detection of low concentration of influenza viruses and COVID-19 by graphene biosensors have been achieved recently. Some works [1,2,7] report the detection of a viral concentration of ~1 fg/mL, which is comparable or below the detectable limits of modern laboratory methods of enzyme-linked immunosorbent assay. However, solving the above-mentioned practical tasks requires the high reproducibility of viral detection.

Reproducibility issues relating to the structural and physicochemical properties of graphene resistances are discussed in [1,7,11,12]. The inhomogeneity of the properties of graphene films over the chip area has led to the need to use several duplicate resistors in one biosensor. This makes it possible to neutralize the effect of inhomogeneity of resistance values on the results of virus detection by the biosensor [1,7,11]. These works show that the hook effect [13] related to a nonlinear dependence of detected signals versus analyte concentration may cause the lack of reproducibility of detection. The nonlinear dependence of detected signal versus analyte concentration is usually observed at both low (2–20 fg/mL) and high (1–10 ng/mL) concentrations [7]. Moreover, this effect can be observed not only in graphene, but also in other 2D materials that have a honeycomb structure [13]. The reasons behind the phenomenon are not yet clear. It would appear that the 3D properties of analyte and structural features of biosensor materials might lead to this effect [13]. In addition, improvements in synthesis, processing, and integration are necessary to implement the large-scale and widespread manufacture of 2D devices for health-related applications [14]. In particular, achieving large-scale uniformity of material properties is essential in order to implement the mass production of biosensors.

The commonly accepted concept of viral biosensor production is based on the creation of conditions for the antibody–antigen (virus) immune reaction on a graphene surface. It can include controlled treatment (functionalization) of the graphene surface to create covalent bonds that ensure the occurrence of selective chemical reactions of the attachment of biomolecules (antibodies) [1,2,3,4,5,6,7,8,9,10]. The antibody–antigen immune reaction on the graphene surface changes its electronic state, which can be registered, for example, by a change in the current flowing through a graphene chip. It is obvious that the ultimate sensitivity of such a biosensor is determined by the properties of graphene as well as by physical and chemical processes that occur on the graphene surface at all stages of the biosensor production. Moreover, it is also affected by the degree of homogeneity of these processes over the entire area of the graphene chip. In [1,2,7,8,9,10], the main attention was paid to both the functionalization of graphene and the design of the contact pads that ensure the detection of viruses on the graphene surface of a chip.

Numerous techniques such as vapor deposition, epitaxial growth, mechanical and chemical exfoliation, and the thermal destruction of the silicon carbide surface have been explored for achieving desired properties in graphene [15]. A short review of these techniques and an analysis of the quality of obtained graphene films can be found elsewhere [1,2,7,15,16,17]. It should be noted that the exfoliation method used by K. Novoselov and A. Geim in their first work on the preparation and study of graphene [7] is reduced to the separation of a one-atom-thick flake from a graphite crystal. Until now, graphene samples obtained by this technology have had the best structural perfection. However, their small size and irregular and unpredictable geometrical shape do not allow for the exfoliation method in industry. Graphene films obtained by thermal destruction of the surface of silicon carbide (SiC) come second in terms of structural perfection [2,17]. Thus, it is possible to obtain structures up to industrially important dimensions. In this case, the dimensions are limited only by the initial SiC substrate, i.e., up to 6 inches (150 mm) in diameter [2,17]. However, this technique also cannot fully avoid the issues of inhomogeneity of graphene quality. Apart from perfect monolayer graphene, there are typical inclusions of bilayer and multilayer graphene [18,19]. Typically, the grown samples are composed of 85% monolayer graphene and 15% bilayer graphene that is represented by small bilayer patches (inclusions) of various sizes. It is reasonable to assume that the unsaturated edges of these inclusions may create extra nucleation ions [18]. More information about the intrinsic properties of the epitaxial graphene on SiC can be extracted from the analysis of the Raman mapping data [19]. Meanwhile, the structural properties of graphene films fabricated by different methods vary significantly. Even for films fabricated by the same method, the nanostructural arrangement of graphene depends on the technological conditions and properties of the substrate material. There are few publications on the influence of graphene nanostructural arrangement on the properties of the chips and biosensors based on them. Its influence on the properties of graphene inclusions is also not discussed enough. The functionalization of graphene is discussed in various publications [1,7,10,20]. Fewer studies have been dedicated to the investigation of changes in the properties of graphene inclusions during functionalization and on the influence of these changes on antibody and virus binding.

In this paper, we used graphene films obtained by thermal destruction of the SiC substrates surface for biosensor chips used to detect the influenza viruses. We show the changes in the resistance, low-frequency noise amplitude, and graphene nano-arrangement reflected in surface morphology during the main stages of biosensor chip development (functionalization of the graphene surface, immobilization of antibodies, detection of influenza viruses). The uniformity of the distribution of these changes over the chip area was also investigated by probe methods of analysis, as well as by a low-frequency noise technique.

## 2. Materials and Methods

The main stages of graphene-based biosensors are presented in Figure 1 and explained below.

### 2.1. The Production of Graphene on a SiC Surface Using the Sublimation Method

In our experiments we used 4H-SiC substrates with a minimum misorientation angle (α ~ 0), and the growth was carried out on the (0001) ± 0.25° orientation (Si face). We used semi-insulating substrates. For the successful development of graphene growth technology, a necessary condition is the high-quality preparation of the SiC surface, which reduces the effect of contamination and surface inhomogeneity on the sublimation process. Pre-growth etching in a hydrogen atmosphere was used for preliminary cleaning of the SiC substrate surface. The essence of the technology lies in the high-temperature heating of the SiC substrate in a hydrogen atmosphere. At high temperatures, free carbon formed on the SiC surface binds with hydrogen to form volatile chemical compounds. We used a gas mixture containing argon (volume fraction 95%) and hydrogen (volume fraction 5%). Then, the growth of graphene on the SiC surface was performed at a temperature of 1700–1800 °C in an argon atmosphere (720–750 torr). The growth process was carried out in a graphite crucible with induction heating.

After the graphene films growth, a conventional photolithography process was used to pattern the graphene/SiC chips. The chips were processed from several samples of graphene films formed by thermal decomposition of semi-insulating 4H-SiC. Details on graphene film processing and the mounting of chips on holders can be found elsewhere [14]. This study was carried out on chips with two contact pads (graphene resistors) assembled on a convenient printed circuit board (PCB) holder. The size of the sensor area (active surface of graphene in the chip) was about 1 × 1.5 mm^2^.

### 2.2. The Functionalization of the Graphene Surface

To provide sensing ability, graphene functionalization is usually accomplished by using various covalent and noncovalent approaches [5,8]. Graphene functionalization modifies the surface chemistry of graphene and creates covalent bonds on its surface which are used to attach a specialized immune protein, an antibody. We used covalent graphene functionalization, as it is the most simple, reliable, and affordable method [2].

In this work, the process of the functionalization of the graphene surface in chips was carried out in two stages: (1) the formation of covalent bonds during the deposition of nitrophenyl groups (nitrobenzene, C_6_H_5_NO_2_) and (2), the subsequent reduction of the nitrophenyl groups to phenylamine groups (aminobenzene, C_6_H_5_NH_2_) by the method of cyclic voltammetry (CV). All CV experiments were performed in a conventional three-electrode cell with an Ag/Ag+ (or Ag/AgCl) reference electrode, a platinum wire counter electrode, and a graphene/SiC chip as the working electrode. The three-electrode cell had a hermetic lid allowing the electrolyte and the space above it to be purged by dry Ar to remove traces of the moisture from the cell and the electrolyte.

At the first CV stage, the nitrophenyl groups were attached to the graphene surface. For this, a graphene chip assembled on a holder was immersed for 1–2 min in a non-aqueous electrolyte based on a mixture of 2 μM 4-nitrobenzenediazonium tetrafluoroborate (4NDT) and 0.1 M tetrabutylammonium tetrafluoroborate (TBATF) in acetonitrile (CH_3_CN).

In the second CV process, the graphene/SiC die was immersed in a 0.1 M KCl water/ethanol (9:1) solution in order to reduce the nitrophenyl groups to the phenylamine groups on the graphene die surface. Details on the graphene functionalization process can be found in [20].

### 2.3. Antibody Immobilization and Influenza Virus Detection

After surface functionalization, all chips were incubated in a solution containing influenza A (or B) antibodies for 3 h at 37 °C, followed by a single wash in PBS. The detection of influenza antigens (viruses) in PBS was then carried out. Concentrations of influenza virus in PBS solutions ranged from 10^−16^ g/mL to 10^−9^ g/mL.

For antibody immobilization on the functionalized graphene surface, we used the same concentration of antibodies diluted in the buffer solution in all experiments. Anti-NP monoclonal antibodies were dissolved at 200 μg/mL in PBS. The concentration of the antibodies was higher than the possible places with covalent bonds suitable for the antibody’s attachment. We did not observe changes in conductivity after the antibody conjugation.

The following strains of influenza viruses used in the experiments were obtained from the collection at the museum of viruses of the Smorodintsev Research Institute of Influenza, Russia: influenza virus A/California/ 07/09 (H1N1pdm09) and influenza virus B/Brisbane/46/15. All experiments with viruses were carried out in a BSL-2 facility at the Institute of Influenza by its employees, who are co-authors of the work. All permits were in place.

Lysates of purified virus concentrates were used as an analyte. The lysates were pre-pared by diluting viruses in a lysis buffer (200 mM DTT, 0.05% Tween 20 in PBS), followed by a freeze-thaw step. Such viral lysates mainly contain destroyed virions. Therefore, the concentration of viruses in the lysates was assessed by measuring the total viral protein with the modified Lowry method, using the RC DC Protein Assay Kit (Bio-Rad, Hercules, CA, USA). Analyte solutions were prepared by tenfold dilution in PBS. The analytes were incubated at room temperature.

The biosensor concept in our studies is based on antigen–antibody immunoreaction on the graphene surface. The selectivity or specificity for sensing performance is mainly due to nature of the immunoreaction: only related (matched) antigens and antibodies participate in the interaction and can change the state of the graphene. However, other viruses can influence the biosensor response via mechanisms other than the immunoreaction.

In this study, we did not use a special passivation of the graphene surface. The response of biosensors with immobilized antibodies was investigated under the conditions of diluted solutions of related antigens in order to determine the influence of the graphene surface on the detection process.

During the detection process, a direct current voltage (20–80 mV) was applied to the graphene chip coated (immersed) with the influenza antibody, and the chip was immersed in a PBS diluted solution of influenza virus (antigen) for 30 s. The influenza antigen chemically attaches to the influenza antibody which results in a change in the resistance of the graphene channel, which can be promptly detected by the passing of a current through the graphene/SiC chip. Thereafter, the chips were pulled out, washed in pure PBS solution, dried, and immersed again in another PBS solution with a different influenza virus concentration.

### 2.4. Methods

Current–voltage (I–U) characteristics and low-frequency noise spectra containing information on the defective system state, which depict the quality of the material, were studied in the chips after each stage of the biosensor fabrication. The surface morphology and the surface potential distribution were monitored by atomic force microscopy (AFM) and Kelvin probe force microscopy (KPFM). In addition, scanning electron microscopy (SEM) was used to visualize the attachment of antibodies and influenza viruses to the graphene surface.

AFM and KPFM measurements were carried out on an Ntegra AURA setup (NT-MDT, Russia). AFM studies were carried out using the HA_FM cantilever (www.tipsnano.com, accessed on 22 December 2021). A resonant mode of operation was used in the work. The AFM probe knocks on the surface scanning frequency 0.6 Hz. The scanning speed was approximately 1.3 μm/s. The stiffness coefficient of such a cantilever is 3.5 N/m, the radius of curvature is less than 10 nm, and the scanning field size is 256 × 256 points.

The I–U characteristics were measured using the KEITHLEY 6487 power source. The power spectral density of voltage fluctuations was measured for the frequency range of 1 Hz to 50 kHz. The studied samples were connected in series with a low-noise load resistor R, the resistance of which varied from 100 Ω to 13 kΩ, depending on the current passing through the chip. The voltage fluctuations S_U_ at the resistors R_L_ were amplified by a low-noise preamplifier SR 560 (Stanford Research Systems, Sunnyvale, CA, USA) and subsequently measured by an SR 770 FET NETWORK Analyzer (Stanford Research Systems, Sunnyvale, CA, USA). The background noise of the preamplifier did not exceed 4 nV/√Hz at 1 kHz, which is approximately equivalent to the Johnson–Nyquist noise of a 1000-Ω resistance. SEM analysis of the chip surface was carried out by a JSM 7001F microscope (Jeol, Tokyo, Japan) in the secondary electron mode at an accelerating voltage of 5 keV and a beam current of 12 pA.

Raman spectroscopy measurements were carried out at room temperature in the backscattering geometry using a T64000 spectrometer (Horiba Jobin-Yvon, Palaiseau, France) equipped with a confocal microscope. The laser power of a YAG:Nd laser with a wavelength of 532 nm was limited to 1.0 mW in a spot 1 μm in diameter to prevent the damaging and modification of the graphene films. Along with local measurements, sample areas of 10 × 10 μm^2^ were analyzed with subsequent plotting of Raman maps of spectral lines parameters. A YAG: Nd solid-state laser with a wavelength of 532 nm was used as an excitation source.

## 3. Results and Discussion

### 3.1. Investigation of the Properties of Graphene Chips before and after Functionalization

The I–U characteristics of all chips under study remained linear at each stage of the biosensor development. Table 1 shows the typical resistance of chips obtained from graphene/SiC plates of different series and the low-frequency noise S_U_ before and after functionalization. The chips from several plates were investigated. The first five characters in the chip notation in Table 1 (like EG319) indicate the plate number on which the graphene film was grown. Characters after a dash indicate the number of the chip processed from this plate (like EG319-3). For each plate, the parameters of a specific chip typical for chips of this plate are given. Chips from different plates are combined into two groups that differ in the percentage of the inclusions of the bilayer graphene into the graphene monolayer. The presence of the inclusions of the bilayer graphene of different sizes is a typical feature of graphene film obtained by thermal decomposition of silicon carbide [16].

The presence of graphene films on the SiC surface was confirmed by Raman spectroscopy, as shown in Figure 2. Before and after functionalization, the Raman spectra of the chips in the region of 1300–3000 cm^−1^ were dominated by sharp G and 2D lines characteristic of monolayer graphene [21] and wide asymmetric bands centered at approximately 1380 and 1550 cm^−1^ corresponding to the buffer layer [22]. After functionalization, a new D line at ~1350 cm^−1^ appeared in the spectra. This is attributed to the appearance of defects in the graphene’s crystal lattice. We did not observe any significant shift or broadening of the G and 2D lines after the functionalization of the chips.

Analysis of the full width at half-maximum of the 2D line (FWHM2D) distribution before and after functionalization allowed us to analyze the distribution of mono- and bilayer graphene areas on the surface of the chips. Figure 2b,c demonstrates the difference in graphene film thickness between samples from Group 1 and 2 before functionalization. In the areas with FWHM2D > 40 cm^−1^, this line has an asymmetric contour corresponding to the envelope of four Lorentzians, which is a fingerprint of bilayer graphene [23]. One can see that the samples from Group 1 have relatively low share of bilayer inclusions (~5%), while in case of samples from Group 2, the share of bilayer inclusions is significantly higher (~30%). After functionalization, the shape and size of bilayer graphene inclusions did not change. In the surface potential maps (Figure 2d), the bilayer inclusions appear as regions of higher potential values.

The low-frequency noise spectra of graphene chips are shown in Figure 3. For all chips, a spectral dependence close to S_U_ ~ 1/f was observed before and after functionalization. This type of dependence is typical in graphene. The dependence indicates that the noise is determined not by uniformly distributed single defects in the material but by a system of defects [24,25]. A higher noise indicates a greater level of defectiveness in the material [26].

The resistance of chips in Group 2 is noticeably lower than that in Group 1 before functionalization (Table 1). An opposite trend in the properties of the chips from Groups 1 and 2 was observed in changes after functionalization. The resistance of the chips in Group 1 decreased significantly. However, S_U_ grew. Meanwhile, the chips from Group 2 showed no significant changes in these parameters. The changes in Raman spectra for the chips in Group 1 are similar to those presented in [7], while there is no noticeable change in the spectra for the chips in Group 2. These results allow us to assume that one of the reasons behind the observed phenomenon is the significant difference in the amount of bilayer graphene inclusions in the chips of these two groups.

We employed AFM to clarify the behavior of the bilayer graphene inclusions. Studying the surface morphology of the graphene chips of these two groups revealed significant differences in the nature of their nano-arrangement both before and after functionalization, as illustrated in Figure 4, Figure 5, Figure 6, Figure 7 and Figure 8. It should be noted that AFM images of the surface morphology were obtained in fields of 25 × 25 μm^2^, 8 × 8 μm^2^, and 2000 × 2000 nm^2^. The most informative in terms of nano-arrangement are the 2000 × 2000 nm^2^ fields. It should be noted that the 2000 × 2000 nm^2^ fields are the most informative in terms of studying nano-arrangement because the chips in Group 1 typically had the smallest bilayer inclusions. 

All the chips exhibited a honeycomb like structure typical of graphene films formed on the Si-face of silicon carbide [27]. Local bright areas of small lateral sizes and heights distributed non-uniformly over the graphene surface were observed on all chips (Figure 4 and Figure 5). These are similar to large inclusions identified as a bilayer graphene in Figure 2 and are only just visible in small AFM scans 2 × 2 µm^2^. The Group 2 chips had a rather higher AFM profile, up to 10–15 nm, and inclusions occupied a larger area (Figure 5) than the Group 1 chips (Figure 6). The height of the AFM profile in the bright areas in Figure 6 is less than 6 nm.

There is a significant decrease in the sizes of bright areas in the chips of Group 2 (Figure 6) and Group 1 (Figure 7) after functionalization. Meanwhile, the sizes of dark regions increase up to 500 nm. The dark regions are similar to shallow pores in plain or large honeycomb-like defects non-uniformly distributed over the chip area. Thus, we can conclude that functionalization changed the nano-arrangement in the graphene, making it less uniform.

AFM profiles depicting the surface of the chips before and after functionalization in Figure 8 visualize these changes and the differences between the features of pristine graphene nano-arrangements in Group 1 and 2 chips. It can be seen that pristine graphene is more defective in chips from Group 2 than in chips from Group 1 (Figure 8a). The density of honeycomb-like defects in graphene in chips from Group 2 is higher. This correlates with higher low-frequency noise in the chips from Group 2 (Table 1).

Functionalization results in the occurrence of honeycomb-like defects with nano- steps (Figure 8b–d), which are deeper in the chips from Group 2 (Figure 8b). We assume that these changes in the graphene nano-arrangement may lead to an increase in conductivity similar to the case when conductivity increases in the process of porous graphene creation [28]. Moreover, in this work, we used a two-stage functionalization process. The first stage was the formation of covalent bonds during the deposition of nitrophenyl groups. The second stage was the subsequent reduction of the nitrophenyl groups to phenylamine groups by a method of cyclic voltammetry. Details on graphene functionalization can be found elsewhere [20].

The covalent binding of nitrophenyls to graphene films is known to lead to a remarkable decrease in conductivity. This happens because of a reduction in graphene aromaticity due to the transformation in hybridization of carbon atoms from sp^2^ to sp^3^. Nitrophenyl groups are acceptors that reduce electronic density in graphene. The attachment of phenylamine groups (aminobenzene, C_6_H_5_NH_2_) by cyclic voltammetry leads to decreasing resistance values, since aminophenyl groups have weaker acceptor properties than nitrophenyl groups.

Thus, graphene nano-arrangement, in addition to its functionalization, can contribute to a decrease in graphene resistance. All the obtained results confirm that functionalization is accompanied by an increase in graphene defectiveness due to the formation of large honeycomb-like defects up to 500 nm in plane. At the same time, small inclusions of bilayer graphene disappear or decrease noticeably. This phenomenon might be related to the reduction properties of phenylamine groups. These results allow us to suggest that shallow bilayer inclusions contain nonequilibrium phases of weakly oxidized graphene. 

SEM images of the surface morphology of a graphene chip after functionalization, immobilization of influenza B antibodies, and antibody–virus B antigen immune reaction are presented in Figure 9. The aggregation of antibodies and antigens and their non-uniform distribution over the graphene surface are observed (Figure 9). These phenomena are going to be discussed later using the results of AFM studies.

Insert 1 in Figure 9 shows that functionalization changes the emission properties of graphene films. This can be identified by a change in contrast between functionalized (dark areas) an unfunctionalized (gray areas) regions in graphene.

The further stages of biosensor fabrication (immobilization of antibodies of influenza A and B viruses, antibody–influenza immune reaction, and detection of influenza viruses) were studied mostly on the chips of Group 1. Significant changes in resistance and low-frequency noise, which are comparable to changes after functionalization, were not observed after these stages. As a result, we chose probe methods to study the chips after each of these stages.

### 3.2. Study of Immobilization of Antibodies of Influenza A and B Viruses, Antibody–Influenza Immune Reaction, and Detection of Influenza Viruses by Biosensors Based on Graphene Chips

The conventionally accepted method described in Section 2.3 was used to detect the influenza viruses (antigens) [2]. Influenza virus antigens were diluted in PBS. In the experiment, the current passing through the chip was measured versus the concentration of antigens of influenza A viruses in PBS in the concentration range of 10^−16^ g/mL to 10^−10^ g/mL. Figure 10 shows an almost monotonic increase in the magnitude of the response, which is approximated by a logarithmic function with the parameter R^2^ close to 1 (0.96) in chips from Group 1 with lower concentration of honeycomb-like defects. A similar dependence was observed when the response of a graphene-based biosensor used to contact with solutions of egg albumin in PBS was studied [20,29]. It should be noted that a weak logarithmic dependence of the detected signal versus analyte concentration, as presented in Figure 10, is closer to linear than ones observed in other studies concerning viral detection [1,7,12]. The reasons behind the weak concentration dependence have yet to be clarified. The biosensor’s response dependence versus the virus’s concentration in chips from Group 2 is strongly nonlinear.

We assumed that aggregation and graphene nano-arrangement might influence the trend of the concentration dependence of detected signal. Therefore, we studied AFM profiles of functionalized graphene surfaces after the immobilization of influenza virus antibodies and after antibody–antigen influenza virus immune reactions. Figure 11 shows alterations in the AFM profile of the chips in accordance with the graphene surface treatment stages. The maximum magnitude of the AFM profile of the Group 1 chips did not exceed 6 nm both before and after functionalization. The profile changes after all stages of graphene surface treatment are shown in Figure 11a (curves 1, 2, and 3). The features of the surface relief after graphene functionalization (curve 1) and after antibody–influenza virus immunoreaction (curve 3) are not indistinguishable on the same axis scales. Therefore, the AFM profiles after these two stages are shown with a larger scale for the y-axis in Figure 11b and with larger scales for both the x- and y-axis in Figure 11c.

Probe methods revealed the aggregation of antibodies and their non-uniform distribution over the graphene surface of the chips (Figure 12); these were also revealed for the antigens (Figure 9 and Figure 13a). For the antibodies, the maximum magnitude of the AFM profile was observed to be in clusters, which means that aggregation occurred both laterally and vertically.

Because of this aggregation, it was difficult to estimate the lateral dimensions of a single antibody. Its vertical dimension was about 15–20 nm, while in clusters it was more than 20–25 nm, as can be seen in Figure 12. A wide variety of sizes of virus aggregates and their inhomogeneous distribution over the chip area can be observed in the SEM images in Figure 9 and Figure 13a, and in the AFM image in Figure 13b. It should be noted that the maximum sizes of aggregates obtained by these two methods correlate well.

The maximum lateral dimension of the antigen aggregates reached 5 μm and their height reached up to 250–300 nm. An enlarged image of one of the aggregates with a lateral dimension up to 5 μm is shown in inset 2 in Figure 9. The appearance of this aggregate is similar to the SEM images of swine flu aggregates found elsewhere [30]. It was difficult to determine the exact lateral and vertical size of a single viral cell under aggregation conditions. However, it can be estimated as 50–100 nm.

The aggregation of antibodies and viruses seems to be one of the reasons for the weak dependence of the current passing through the chip (chip response) on the concentration of viruses in PBS solutions, as shown in Figure 10. The mechanism leading to the formation of aggregates requires additional research. Understanding this mechanism is especially important for the reliable detection of low concentrations of viruses at early stages of infection.

When the viral concentration is high and it is necessary to register the presence of the virus, aggregation becomes a positive factor. Recently, a hexagonal honeycomb-like structure was created artificially by etching or laser-engraving [28]. Studies of aggregates by probe methods have shown that their formation is often associated with the peculiarities of the honeycomb-like defects. The 2000 × 2000 nm^2^ AFM scan images in Figure 9 reveal the features of this structure, which become apparent in the alternation of honeycomb-like defects whose sizes vary from 100 nm to 500 nm and light regions whose sizes are slightly larger than those of the honeycomb-like defects.

The shape of defects can be hexagonal. At the same time, light ball-like aggregates of antibodies are located either in the dark areas or at the borders of the light regions, i.e., on the nano-step (Figure 12a).

Inset 2 in Figure 9 shows a contrast SEM image of virus aggregates. It can be seen that the viruses are located in a nearly hexagonal-shaped deepening. Figure 14 shows SEM images of virus aggregates smaller than those in Figure 9 but also located in the hexagonal-shaped deepening. Thus, the features of graphene nano-arrangement may be one of the reasons for the aggregation of viruses and antibodies.

## 4. Conclusions

The study has shown that a graphene monolayer with 5% of bilayer inclusions, grown on SiC substrates, could be the basis for creating influenza virus biosensors. A set of diagnostic methods that includes SEM, AFM, Raman spectroscopy, and low-frequency noise measurements allow one to characterize the properties of graphene chips during the main stages of development. We found significant increases in the number of defects as well as a growth in conductivity in graphene chips after functionalization. We associate the observed phenomena with both the attachment of aminophenil groups and changes in nano-arrangement of the graphene. The changes in the graphene nano-arrangement are reflected, in particular, in the features of the surface morphology. Functionalization is accompanied by the formation of large honeycomb-like defects up to 500 nm in plane. At the same time, small inclusions of bilayer graphene disappear or decrease noticeably. An increase in the amount of bilayer or multilayer inclusions in graphene leads to a noticeable growth in the number of honeycomb-like defects. SEM and AFM measurements revealed that these defects facilitate the aggregation of antibodies and influenza viruses. Furthermore, they depict a non-uniform distribution of aggregated antibodies and influenza viruses over functionalized graphene films.

The features of graphene nano-arrangement affect the reliability of detecting extremely low concentrations of viruses during the early stages of diseases. They may also be the cause of observed weak non-linear logarithmic dependence of biosensor response versus the virus’s concentration. A decrease in the concentration of honeycomb-like defects in graphene made it possible to achieve an almost linear-logarithmic dependence of the response of the graphene biosensor versus the virus concentration in the range from 10^−16^ g/mL to 10^−10^ g/mL in PBS.

Thus, the control of graphene nano-arrangement is important to reduce the effect of viral aggregation and to create extremely sensitive biosensors for influenza viruses.

## Figures and Tables

**Figure 1 biosensors-12-00008-f001:**
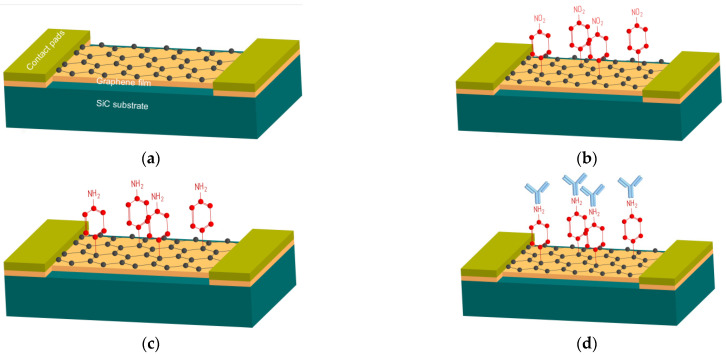
The functionalization of a biosensor based on graphene film on SiC substrate for influenza virus detection. (**a**) Chip with pristine graphene. (**b**) Formation of covalent bonds after deposition of nitrophenyl groups. (**c**) Reduction of the nitrophenyl groups to phenylamine groups. (**d**) Immobilization of influenza A (or B) virus antibodies.

**Figure 2 biosensors-12-00008-f002:**
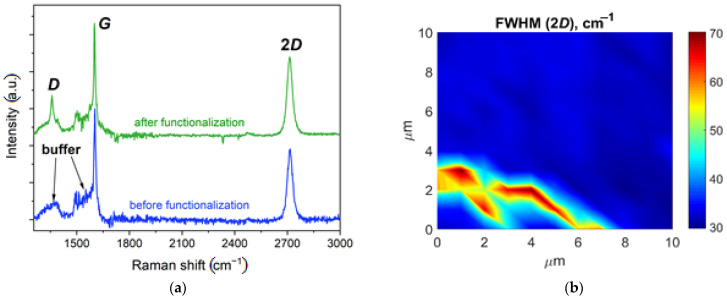
Raman spectra of Group 1 and 2 chips before and after functionalization. (**a**) Typical Raman spectra of monolayer graphene areas in the chip before (blue line) and after (green line) functionalization. The spectra are presented after subtraction of the 4H-SiC substrate spectrum contribution. (**b**) Typical Raman maps of 2D line FWHM (full width at half-maximum) distribution for the chips from Group 1 with ~5% of bilayer inclusions. (**c**) Typical Raman maps of 2D line FWHM distribution for the chips of Group 2 with ~30% of bilayer inclusions. (**d**) A typical surface potential map of a chip from Group 2, which reveals the presence of bilayer inclusions as light spots of different sizes.

**Figure 3 biosensors-12-00008-f003:**
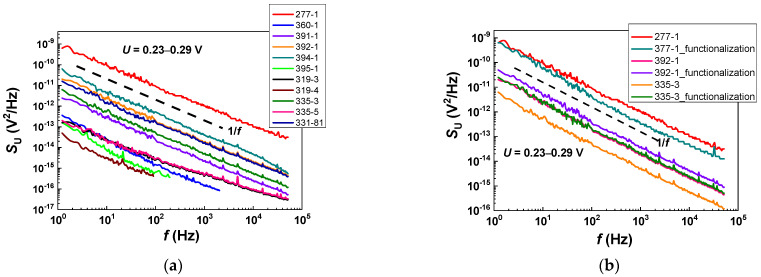
Low-frequency noise spectra of graphene chips before (**a**) and after (**b**) functionalization. Dashed lines indicate a simulation of the 1/f dependence for references.

**Figure 4 biosensors-12-00008-f004:**
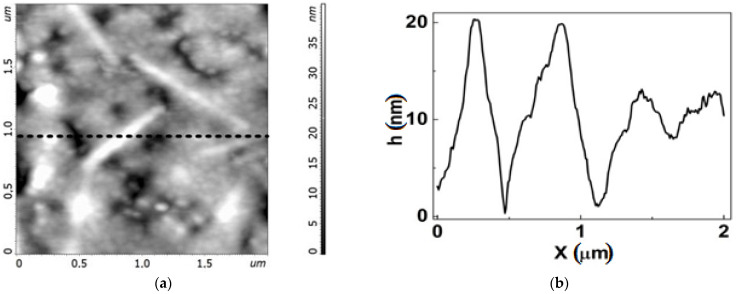
Topography of the graphene surface for chip EG392-1 from Group 2 before functionalization. (**a**) AFM image of the surface with bright areas. (**b**) AFM profile of the graphene surface along the dashed line in (**a**), showing the characteristic arrangement of the nanomaterial of the surface. Bright areas are equivalent to peaks with large magnitude on the profile curve.

**Figure 5 biosensors-12-00008-f005:**
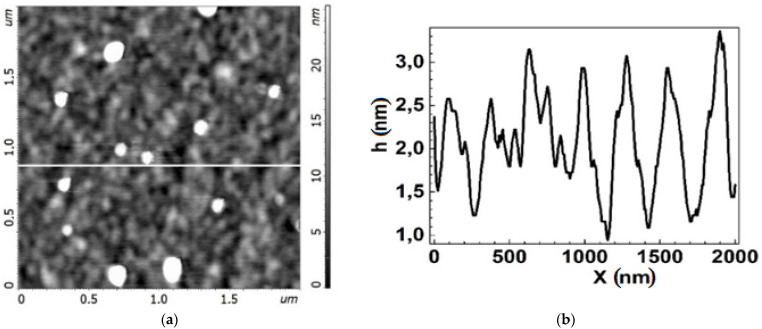
Topography of the graphene surface for chip EG335-5 from Group 1 before the functionalization process: (**a**) AFM image of the surface with less bright areas than in Figure 4a. (**b**) AFM profile of the graphene surface along the white line in Figure 5a, showing the characteristic arrangement of the nanomaterial of the surface. Bright areas are equivalent to peaks with large magnitude on the profile curve.

**Figure 6 biosensors-12-00008-f006:**
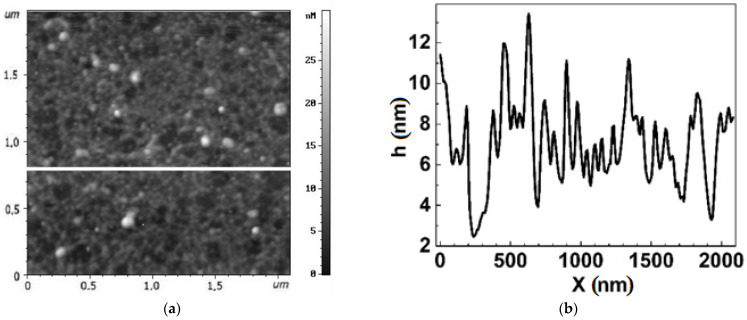
Topography of the graphene surface in chip EG392-1 from Group 2 after functionalization. (**a**) AFM image of the surface. (**b**) AFM profile of the graphene surface along the line in (**a**), showing the characteristic nanomaterial arrangement of the surface. Bright areas are equivalent to peaks with large magnitude on the profile curve.

**Figure 7 biosensors-12-00008-f007:**
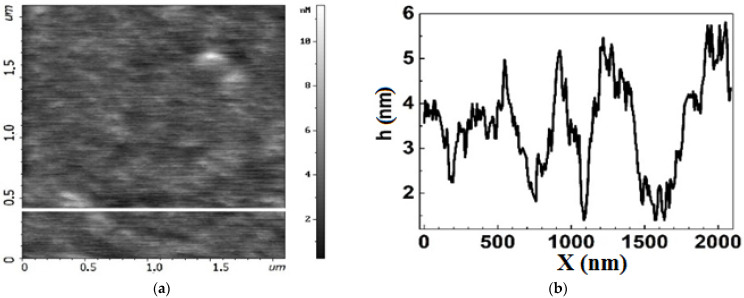
Topography of the graphene surface in chip EG335-5 from Group 1 after the functionalization process. (**a**) AFM image of the surface. (**b**) AFM profile of the graphene surface along the line in (**a**), showing the characteristic nano-arrangement of the surface. Bright areas are equivalent to peaks with large magnitude on the profile curve.

**Figure 8 biosensors-12-00008-f008:**
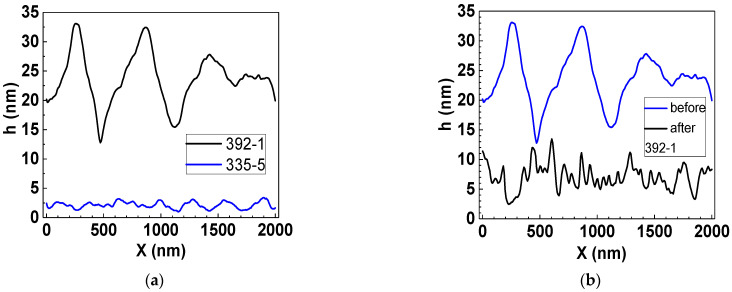
AFM surface profiles of chips before and after functionalization. (**a**) Group 1 and 2 chips before functionalization (pristine graphene in the chips). (**b**) A Group 1 chip before and after functionalization. (**c**) A Group 2 chip before and after functionalization. (**d**) Group 1 and 2 chips after functionalization.

**Figure 9 biosensors-12-00008-f009:**
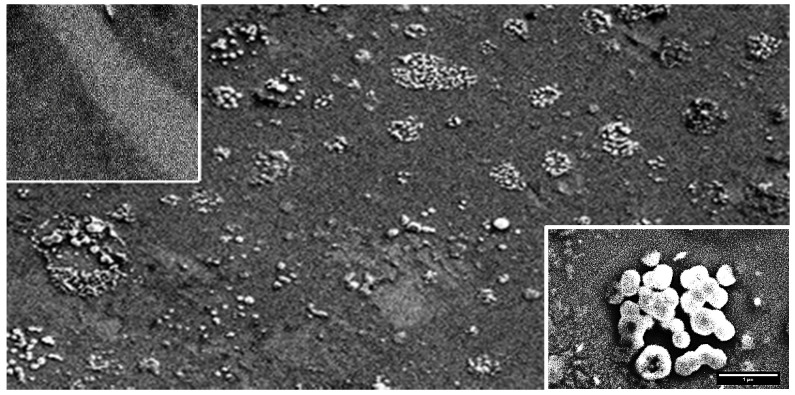
SEM image of a 45 µm × 150 µm graphene chip from Group 1. Inset 1 (**left**, **top**) shows a 5 µm × 5 µm section with boundaries of functionalized graphene (dark areas) and non-functionalized graphene (light area in the middle). Inset 2 (**right**, **bottom**) shows a 2.7 µm × 4.2 µm region with virus aggregates, located in a recess with a facet close to a hexagonal structure.

**Figure 10 biosensors-12-00008-f010:**
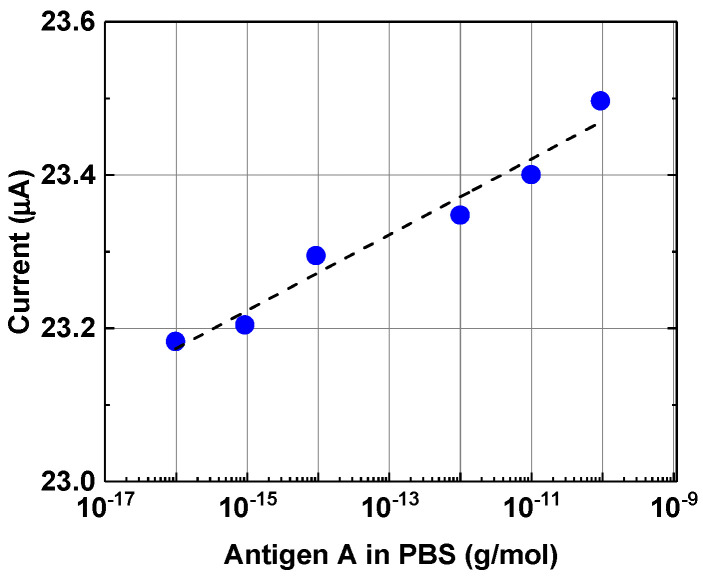
Chip response versus concentration of the virus antigen in the PBS solution. The dotted line represents the approximation of experimental data with a logarithmic function (y = 0.0219 ln(x) + 23.978 R² = 0.9628). Chip EG335-5 from Group 1.

**Figure 11 biosensors-12-00008-f011:**
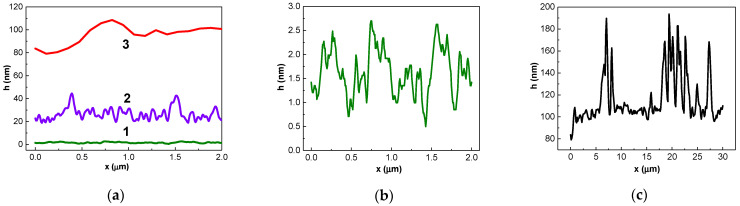
AFM profiles (scan of 2000 nm × 2000 nm) of the chip in accordance with the graphene surface treatment stages. (**a**) Profiles of the graphene surface treatment at different stages: curve 1—pristine graphene surface (stage 1); curve 2—after functionalization and immobilization of influenza B antibodies on the graphene surface (stage 2); curve 3—after functionalization, immobilization of influenza virus antibodies, and antibody–antigen virus B immunoreaction (stage 3). (**b**) Profile of the relief after stage 1 with a larger y-axis scale. (**c**) Profile of the relief after stage 3 with larger x- and y-axis scales.

**Figure 12 biosensors-12-00008-f012:**
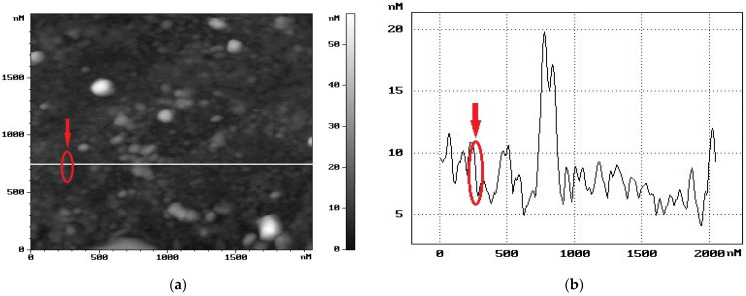
Topography of the graphene surface of a chip from Group 1 after graphene functionalization and immobilization of influenza B virus antibodies. (**a**) AFM image of the surface. (**b**) AFM profile of the graphene surface along the white line in (**a**). The peak in the graph reveals an aggregate on the graphene surface up to a 20-nm height. The arrow denotes a step on the graphene surface.

**Figure 13 biosensors-12-00008-f013:**
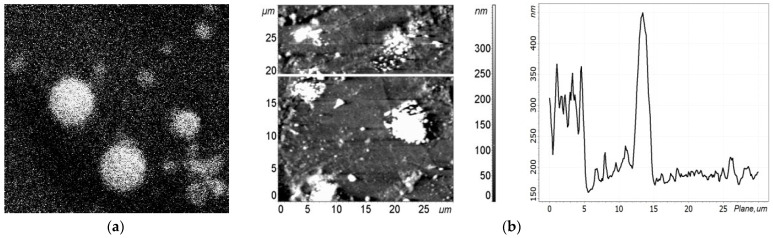
SEM and AFM images of the graphene surface of the same Group 1 chip as in Figure 9 with aggregates of influenza B antigen viruses. (**a**) SEM image of a 5 × 5 μm^2^ surface. (**b**) AFM images. On the (**left**)—graphene surface. On the (**right**)—profile of the graphene surface along the line in the figure on the (**left**). The peaks in the graph reveal the aggregates on the graphene surface with a height up to 250–350 nm.

**Figure 14 biosensors-12-00008-f014:**
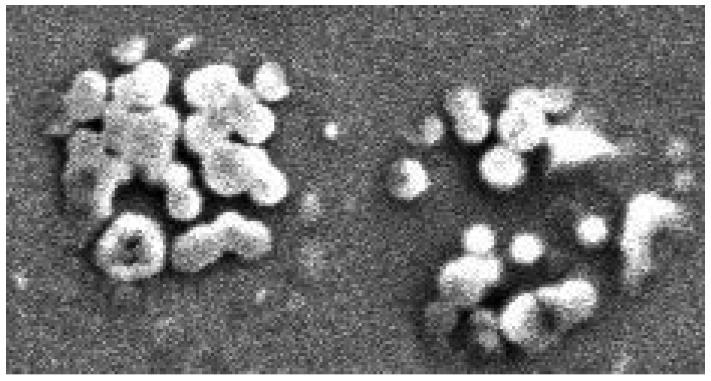
SEM image of aggregates of influenza B antigen (virus) on the graphene surface of a chip from Group 1. Field of view is 2 × 1 µm^2^.

**Table 1 biosensors-12-00008-t001:** Typical parameters of graphene chips before and after functionalization.

Graphene Chip ID		R and S_U_ before Functionalization		R and S_U_ after Functionalization	
		R, Ω	S_U_, V^2^/Hz	R, Ω	S_U_, V^2^/Hz
Group 1	EG319-3	6582	2.00 × 10^−13^	1201	7.94 × 10^−13^
	EG319-4	5718	6.31 × 10^−14^	1178	1.01 × 10^−12^
	EG335-3	5670	6.31 × 10^−12^	1676	2.51 × 10^−11^
	EG335-5	3465	5.01 × 10^−13^	1650	1.26 × 10^−12^
	EG360-1	1691	3.62 × 10^−13^	1233	1.58 × 10^−12^
	EG391-1	2672	2.44 × 10^−12^	2286	8.09 × 10^−12^
	EG394-1	2134	6.31 × 10^−11^	1858	7.94 × 10^−11^
	EG395-1	3251	1.58 × 10^−13^	2566	5.01 × 10^−12^
Group 2	EG392-1	1271	2.04 × 10^−11^	2646	5.01 × 10^−11^
	EG277-1	1445	6.31 × 10^−10^	1713	6.31 × 10^−10^

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
