# Peer review of "Investigation of the Morphology and Electrical Properties of Graphene Used in the Development of Biosensors for Detection of Influenza Viruses"

_biosensors, 2021, doi:10.3390/bios12010008_

Round 1

Reviewer 1 Report

In this work, Shmidt and coworkers presented an investigation on changes in the resistance, low-frequency noise level, and 16 surface morphology of graphene during development of biosensor chips based on graphene films on SiC substrates for the detection of influenza viruses. By carefully reading, I found the manuscript not well organized with overclaimed and suspicious interpretations of the results, and the following should be addressed before the manuscript can be considered further.

  1. There are many typos in the manuscript and should be revised before submission. For example, at line 91 on page 2, “nitrobenzene, C6H5NO2”, the numbers should be subscript to the elements.
  2. Technical terms, such as AFM and SEM should be defined before use.
  3. There should be many investigations on fabrication of biosensor for detection of virus. Further to these, there should also be many investigations on the functionalization and application of graphene for various sensor applications. It would be interesting if related references and results can be cited and compared together.
  4. It would be interesting if the design philosophy for the biosensor investigated in this work can be illustrated in a well designed schematic figure.
  5. The current methods and materials section should be revised to include at least typical procedures for fabrication and functionalization of the chips. This would be one of the major drawbacks.
  6. Raman characterization of the chip before and after surface functionalization should be provided. In addition to raman characterization, other surface characterization would be also necessary.
  7. According to Table 1, there is an abrupt decrease of R after functionalization, it would be necessary to explain the physics behind. At least, a citation should be made. The nomenclature of the samples mentioned in Table 1 should also be explained.
  8. FT-IR or raman spectra would be helpful to root the discussions and proposals on the surface species starting from lines 166. The authors are suggested to find alternative convincing proof to root their discussions. It is also confusing considering the size of the molecules used for functionalization and the observed AFM profiles.
  9. Another major draw back of the current manuscript is that the mechanisms behind the observed or proposed features were not well discussed.
  10. Permit for use of virus in the research disclosed should be obtained and submitted together with the manuscript.

Author Response

Dear Reviewer,

we appreciate very much Your important and deep remarks to our first version of the manuscript. they demanded from us substantial reworking of the material, adding links and attracting additional dimensions. Now the article looks much more complete and, we hope, has become more useful for readers.

Reviewer #1 has 10 comments.

For each comment received, there are responses from the authors. Please find them in the attached file.

Reviewer 2 Report

Comments to the Author

The authors prepared graphene based biosensing platform. The authors presented interest work but still remain some problems for publication. So, I would suggest an acceptance for publication in Biosensors after revision which is suggested as follows:

  1. Regarding the table 1, What is the major difference between group 1 and 2? Is functional group of graphene different?

And could you explain why resistance is changed after functionalization? As authors mentioned, their resistance is tuned and opposite behavior was shown by surface modification. But its reason was not clearly described.

  1. The authors mentioned graphene was thermally deposited on the substrate. So, please inform the size and number of layers of graphene. As the authors knew, number of layers of graphene is related with resistance so that information might be important.

It could be characterized with Raman and so on.

  1. The authors showed several AFM images to characterized graphene-based structures. But there is no scan condition information. So please add scan rate, scan line and so on.

  1. The AFM images from Fig 2 to Fig 4, lots of small particles are depicted. It seemed lots of nanoparticles not graphene is existed on the surface. So, could you explain what is this?

  1. The AFM image of pre-functionalized EG335-5 was exhibited in Fig 3., and post-functionalized chip EG392-1 was shown in Fig 4.. It seems the author would like to claim the difference between pre and post functionalization using these images but actually, it is too hard to find the difference.

In fig 3 and 4 both images, similar sizes of a few large particles and lots of dispersed small size particles are shown. And cross section profile also seems to be similar. Of course, the scale of x axis is different so it looks there are big different but if x axis in Fig 3 (b) is elongated like Fig 4 (b) or the opposite case, these two profiles may look similar.

So, it is hard to catch the exhibition of purpose of these results. 

Please give more clearly and solid decryption

  1. The authors showed conductivity depending on the virus concentrations. But bare graphene and antibody conjugated graphene conductivity also should be compared. So, please show the conductivity changing before and after antibody conjugation.

  1. Please show the selectivity with other viruses for sensing performance.

Author Response

Dear Reviewer,

we appreciate very much Your important and deep remarks to our first version of the manuscript. they demanded from us substantial reworking of the material, adding links and attracting additional dimensions. Now the article looks much more complete and, we hope, has become more useful for readers.

For each of 7 comments received, there are responses from the authors. Please, find them in the attached file.

Round 2

Reviewer 1 Report

In this revision, the authors have made revisions and addressed some concerns of the reviewers, the manuscript looks much better than the original version. However, I have a further concern with the response selectivity of the corrent design. The authors may comment this at least in their conclusions.  

Reviewer 2 Report

Dear. Authors, 

Thanks for revising the manuscript and I agree to accept this article for publication in biosensors.